# Do Hot Executive Functions Relate to BMI and Body Composition in School Age Children?

**DOI:** 10.3390/brainsci11060780

**Published:** 2021-06-12

**Authors:** Paula Szcześniewska, Tomasz Hanć, Ewa Bryl, Agata Dutkiewicz, Aneta R. Borkowska, Elżbieta Paszyńska, Agnieszka Słopień, Monika Dmitrzak-Węglarz

**Affiliations:** 1Institute of Biology and Human Evolution, Faculty of Biology, Adam Mickiewicz University, 61-614 Poznan, Poland; tomekh@amu.edu.pl (T.H.); ewa.bryl@amu.edu.pl (E.B.); 2Department of Child and Adolescent Psychiatry, Poznan University of Medical Sciences, 60-572 Poznan, Poland; adutkiewicz@ump.edu.pl (A.D.); agaslopien@ump.edu.pl (A.S.); 3Faculty of Education and Psychology, Maria Curie-Sklodowska University, 20-400 Lublin, Poland; aneta.borkowska@autograf.pl; 4Department of Integrated Dentistry, Poznan University of Medical Sciences, 60-812 Poznan, Poland; ela@pa.pl; 5Psychiatric Genetics Unit, Department of Psychiatry, Poznan University of Medical Sciences, 60-806 Poznan, Poland; mweglarz@ump.edu.pl

**Keywords:** hot executive functions, body composition, obesity

## Abstract

Deficits of ‘hot’ executive functions (EFs) involving emotional and motivational processes are considered as a risk factor for excessive weight, but few studies have tested the relationship between hot EFs and body composition in children. The aim of the study was to assess the association of the ability to delay gratification and affective decision-making with the body mass index (BMI) and body composition in children with typical neurocognitive development. The sample consisted of 553 Polish children aged between 6–12 y. The delay of gratification task (DGT) was applied to assess the ability to delay gratification. The Hungry Donkey test (HDT) was applied to assess affective decision-making. The indicators of decision-making in the HDT were net score and learning rate. The relationships between hot EFs and BMI, fat mass index (FMI), lean body mass index (LBMI) were tested. The association of the *z* scores of BMI and FMI, overweight/obesity, and the ability to delay gratification was found insignificant after controlling cofounding factors. Most of the results on affective decision-making and *z* scores for BMI, FMI and LBMI were insignificant as well. The relationship between the ability to delay gratification, affective decision-making, and adiposity is not pronounced in typically developed children.

## 1. Introduction

Childhood obesity has become one of the most serious worldwide health challenges [1]. Obesity leads to negative health issues [2], in particular, the increased risk of civilization diseases (diabetes type 2, hypertension, coronary heart disease, cancer) related to high treatment costs [3]. For this reason, in recent years, global actions have been particularly focused on the prevention of excess weight [4]. It is undeniable that the main cause of obesity is energy imbalance, to which excessive food consumption and sedentary lifestyle contribute [5]. However, the reasons why certain individuals have a greater tendency to overeat and exceed their energy requirements still remain unclear [6].

Deficits in executive functions (EFs) are considered to be one of the factors that can affect the development of overweight and obesity. Executive functions are a term for higher-level cognitive processes that allow for goal-oriented behaviors [7]. In other words, EF skills are neurocognitive processes involving the conscious control of thought, emotion and action [8]. It has been noted that some cognitive processes related to EFs may be driven by conscious and unconscious representations of the motivational and affective stimulus [9]. Thus, the literature distinguishes between cold and hot executive functions [8]. Cold EFs are associated with processes involving logic and critical analyses without an affective component, whereas hot EFs involve the regulation of affect and motivation [10]. The core EFs classified as cold are working memory, inhibitory control, and cognitive flexibility [7]. The hot EFs are skills referring to goal-oriented modulation of approach/avoidance evaluation [11]. There is considerable neural and behavioral evidence that the concept of EF skills may be placed on a continuum, from emotionally and motivationally significant context (hot EFs) to emotionally neutral context (cool EFs) [11]. The hot context is triggered during strategy games involving reward perspective (such as snacks, toys, or gifts) and when there is a need to delay the gratification of a tempting reward [12]. It is also revealed in the context of appetite self-regulation. Laboratory test assessing EFs performance can measure both cold and hot EFs. What makes EFs hot is the requirement for a flexible assessment of whether to approach the stimulus or to avoid it [11]. Thus, both cool and hot EFs are involved in self-regulation processes (e.g., inhibitory control); however, hot EFs occur in a motivationally and emotionally significant context [8,11]. The focus of our particular interest is on hot EFs.

According to obesity researchers, EF deficits may partially explain some aspects of eating beyond the need for calories [13]. Some of the studies suggest that EFs deficits are generally related to the mechanisms of food intake, e.g., excessive energy intake [14,15], excessive fat consumption [16], snack intake [17], and emotional eating [16,18]. The problem with impulsivity in obesity can be considered in light of a dual process perspective [19]. Poor self-regulation may be the outcome of two distinct but complementary information processing systems: the automatic (bottom-up) and the reflective (top-down) systems [20]. A decision to go for the immediate reward (desirable food) or to resist temptation for future benefit is the result of a balance between top-down and bottom-up processes. Strong automatic (affective and emotional response to the appetitive stimulus) and weak regulatory processing (executive control) can contribute to overeating in children and, in turn, lead to obesity [19]. According to Appelhans [21], the temptation management approaches different strategies, depending on the cold or hot state. As temptation management represents an aspect of self-regulation, it depends heavily on executive EFs. Thus, the affective decision-making and the ability to delay gratification are EFs in our particular interest.

Several studies have found an association between deficits in the ability to delay gratification and affective decision-making with higher body mass index (BMI) and overweight or obesity [22,23,24,25,26,27]. Other studies, in turn, do not reveal such a relationship [15,28,29,30]. Furthermore, only a few studies have so far tested the relationship between affective decision-making and delaying gratification with body composition in children [31]. It is possible that a body composition analysis may reveal an EF–obesity related tendency in children with normal BMI but abnormal FMI or LBMI. Inclusion of the FMI and LBMI assessment in the analysis can provide an improvement on the use of BMI alone, as the normal BMI may not reveal abnormality in adiposity or lean body mass [32]. We hypothesized that potential EFs may be associated with abnormality in proportions of body composition before overweight or obesity occurs, as they lead to over-consumption of processed food and a sedentary lifestyle. These, in turn, are associated with adiposity [33]. Thus, this method could be suitable for a study of the EF–obesity relationship in children.

The aim of our study was to assess the association of affective decision-making and the ability to delay gratification with BMI and body composition in children with typical neurocognitive development. We decided to exclude children with mental illness diagnoses for two reasons. First, we wanted to examine the relationship between EFs and body composition in typically developed children with no clinically relevant cognitive impairments characteristic of disorder, e.g., ADHD, depression, schizophrenia, that are associated with weight gain [34]. Second, psychopharmacotherapy is often used during treatment. Previous studies have shown that psychotropic drugs can affect body mass [35].

## 2. Materials and Methods

### 2.1. Participants and Procedure

Recruitment for the study was conducted among children between 6 and 12 years of age from primary schools in Poznań, Poland. The criterion for inclusion in the study was the absence of any formal diagnosis of mental disorders in children declared by their legal guardians. Furthermore, the exclusion criteria were organic CNS dysfunctions (e.g., epilepsy, tumors), endocrine disease (e.g., hypothyroidism, Cushing’s syndrome, growth hormone deficiency), and current pharmacotherapy, in order to limit the influence of additional biological and medical variables. Both parents and children were fully informed about the research procedures. All data collection took place in schools and was scheduled between 8 a.m. and 12 p.m. The data on hot EFs, body height and weight measurements, body composition analysis, and socioeconomic factors were collected for each child.

### 2.2. The Selection Procedure

Consent for the research was obtained for 553 children, however, 13 individuals resigned from the study for unknown reasons. Subject characteristics with descriptive statistics are detailed in Table 1 and Table 2. It was not possible to collect a full set of data for all participants. There were six incomplete anthropometric measurements (missing data in body weight or body composition analysis). Moreover, not all parents answered the questions related to their socioeconomic status. DGT was completed by 533 children and HDT was completed in the full version of 200 Trials by 507 children. We decided not to exclude the participants with incomplete data in order to keep as large a group as possible in the analysis.

### 2.3. Hot Executive Functions Assessment

#### 2.3.1. Delay of Gratification Task (DGT)

The DGT is a task that assesses affective self-regulation in children, developed by Mischel and Ebbesen in the 1970s [36]. A child was given a choice between a small reward immediately or a larger one later on. The reward in our study was an egg-surprise (a chocolate egg with a small toy inside). The egg-surprise was placed in front of a child. Next, the child was told that she/he could take it, eat the chocolate, and have a toy or wait for an adult who would bring an additional reward. The researcher then left the room and the child was left alone, observed by a camera. The wait time was 7 min. In every minute during waiting time, the child could take the egg-surprise. The main indicator of affective self-regulation was the eating or unpacking of the egg-surprise by the child. We adopted the time criterion after Hughes et al. [29].

#### 2.3.2. Hungry Donkey Test (HDT)

We adopted a computer-based tool developed to assess affective decision-making in children that is an age-appropriate version of the Iowa gambling task [9]. Children were told to help a hungry donkey collect as many apples as possible by pressing one of four possible buttons (*a*, *s*, *k*, *l*), all of which opened one door (A, B, C, D) on the computer screen. The doors were presented in a horizontal row. After opening the door an outcome with the number of apples gained and/or lost displayed. The wins were higher at doors A and B (advantageous doors) and lower at doors C and D (disadvantageous doors). However, the punishment frequency (losses) was higher for doors A and C and lower for doors B and D. The typical version of the game consists of 100 trials [37]; however, we administered the 200-trial version of the HDT to the participants. Several HDT studies have shown that the 200-trial version better differentiates results among individuals than the original 100-trial version because of the increased possibility to learn and reveal a stable game strategy [38]. A strong preference of a participant to one door was considered if the child response proportion for this door was at least 0.5 and the preference for that door was 0.25 greater than for other doors. The main result and indicator of affective decision-making, net score, was the difference between advantageous and disadvantageous choices [(C + D) − (A + B)] expressed in proportion of choices [38]. An impaired net score was considered when the results of a child were ≤0.0. The next indicator was learning rate. A plus learning rate was understood as the improvement of performance in Trials 101–200 compared to Trials 1–100. The improvement was considered when the net score in Trials 101–200 was higher than the net score in Trials 1–100.

### 2.4. Anthropometric Assessment and Excess Weight Diagnosis

Body weight measurement and body composition analysis (body fat, fat free mass) were carried out with the use of the multi frequency segmental body composition analyzer TANITA MC-780. Body height was measured using the anthropometer Seca 213, with a measurement accuracy of ±1 mm. BMI was calculated on the basis of the height and weight measurement. The fat mass index (FMI) [32] was calculated on the basis of the height and body fat measurement. The lean body mass index (LBMI) [32] was calculated on the basis of the height and fat free mass measurement. The BMI was adjusted for sex and age on the basis of WHO child growth standards [39] with the use of WHO AnthroPlus software [40]. Underweight, healthy weight, overweight, and obesity were diagnosed with the International Obesity Task-Force criteria [41,42] based on children’s BMI. The LBMI and FMI were adjusted for sex and age based on the data of the sample. The description statistics of the whole sample are presented in Table A1, Table A2 and Table A3 in Appendix A.

### 2.5. Controlled Factors

The parents of the participants filled in a questionnaire in order to assess the socioeconomic status of the family. The questionnaire consisted of questions about the level of the parents’ education (based on their formal degree of education: primary, vocational, secondary, or higher education), parental subjective assessment of their socioeconomic situation (5-point scale from ‘very bad’ to ‘very good’), and the size of place of residence (six categories from ‘village’ to ‘cities with over 500 thousand residents’). The effects of these variables, as well as of sex and age, were controlled in analyses because of their possible role as moderators in the association of EFs and BMI/body fat.

### 2.6. Statistical Analysis

All the analyses were performed using IBM SPSS Statistics 25.0. All statistical tests were two-tailed and computed with a significance value of *p* < 0.05.

#### 2.6.1. The Ability to Delay Gratification

To assess the differences in the *z* scores for BMI, FMI, or LBMI between children who succeeded and failed in DGT, the independent-sample-*t*-test was applied. To assess the differences in the DGT failure frequencies between children with excess weight (overweight + obesity) and healthy weight the χ^2^ test and logistic regression analysis were applied. The effect of possible confounding factors (SES, age, mother’s BMI) on the excess weight was controlled in the adjusted logistic regression analysis of the DGT result and BMI status association.

#### 2.6.2. Affective Decision-Making—Net Score

To assess the differences in BMI, FMI, and LBMI *z* scores in children with an impaired net score and a positive outcome in HDT, the independent-*t*-test was applied. The r Pearson correlation was calculated to assess the association of the net scores at Trial 1–100, Trial 101–200, and throughout the whole game with body mass and composition indicators. Next, for the significant correlation of net score and BMI, the multiple linear regression was applied, including cofounding factors (SES, mother’s BMI). To control our analysis of the net score–BMI/body composition for age, we used age and sex adjusted *z* scores for BMI, FMI, and LBMI. Furthermore, we included age as a covariate in linear regression analysis.

#### 2.6.3. Affective Decision-Making—Learning Rate

The independent-sample-*t*-test was applied for testing the differences between children’s BMI, FMI, and LBMI *z* scores according to their learning rate status (*plus* or below 0.0). In order to evaluate children’s improvement performance with the duration of the task, a mixed-design ANCOVA was applied, with control of confounding factors (sex, age, SES). The sphericity assumption was tested with Mauchly’s test. Since the sphericity criterion was not met, the lower-bound correction was applied. The within-subject factor was learning curve, understood as the score in 10 consecutive blocks of trials.

#### 2.6.4. Affective Decision-Making—Door Preference

In order to evaluate children’s door preferences in the whole sample, description statistics were applied. The next step was to assess the association of door preferences with BMI status by applying χ^2^ test.

## 3. Results

### 3.1. The Ability to Delay Gratification 

The independent-sample *t*-test did not reveal any differences in the *z* scores for BMI, FMI, or LBMI between children who succeeded and failed in the DGT (Table 1). Although these body weight status indicators did not differ between the groups, the χ^2^ test showed that excess body mass (overweight + obesity) was related to an increased risk of failure in DGT in comparison to children with a healthy weight (χ^2^ = 4.81, OR = 2.62, *p* = 0.036). Nevertheless, the association did not remain significant in the logistic regression analysis adjusted for controlled factors (OR = 0.43, *p* = 0.076). The detailed results of logistic regression are presented in Table 2.

### 3.2. Affective Decision-Making

#### 3.2.1. Net Score

The statistical analysis revealed significantly higher BMI and FMI *z* scores in children with a net score >0.0 in comparison to children with results below 0.0 (Table 1); however, the effect size measured by the Cohen’s *d* was small (*d* = 0.20) and very small (*d* = 0.19) for the *z* score BMI and FMI, respectively. The r Pearson correlation revealed very poor but significant association of the entire game net score with BMI (*p* < 0.05). This association lost statistical significance after controlling cofounding factors in the multiple linear regression, such as socioeconomic status and mother’s BMI (Table 3).

#### 3.2.2. Learning Rate

Statistical analyses did not reveal any differences between children’s BMI, FMI, and LBMI *z* scores according to their learning rate status (Table 1). Children who improved their performance in Trials 101–200 had statistically the same body mass and composition indicators values as children who did not improve.

The analysis of ANCOVA showed a significant improvement in all children’s performance with the duration of the task (*p* = 0.005). However, no interaction between learning and BMI status was observed (*p* = 0.161). There were no significant differences in learning between children with different BMI status (*p* = 0.130). The mean scores by block of trials according to body mass status are shown in Figure 1.

#### 3.2.3. Door Preference

Analyses of the whole sample results with the use of the dependent *t*-test revealed that during Trials 101–200, children were significantly more likely to choose the advantageous doors in comparison to Trials 1–100 (*t* = −6.595, *p* < 0.001). Moreover, only 11.4% of children met the preference criterion for a single door: 5.1% preferred door B (disadvantageous door, high gain, low punishment frequency) and 6.3% preferred door D (advantageous door, low gain, low punishment frequency). These preferences were not related to BMI status (χ^2^ = 10.286, *p* = 0.591). The mean proportion of choices by door for each 100-trials in the whole sample are shown in Figure 2.

## 4. Discussion

The aim of our study was to verify if the ability to delay gratification and affective decision-making are related to BMI and body composition of typically developed children of early-and-middle-school age. The prevalence of worldwide overweight and obesity (over 41 million children aged 5–19 (WHO, 2019)) indicates the need to explore all potential factors that may be relevant for obesity prevention and treatment. In our sample, the frequency of excess body mass was also high—21% (15.8% overweight, 5.2% obesity). We hypothesized that the association of affective decision-making and the ability to delay gratification may be unrelated to BMI but associated with body fat and lean body mass. The use of indicators that take into account the proportion of fat and muscles to body height (FMI and LBMI) constituted an additional value of the study.

### 4.1. The Ability to Delay Gratification

According to prior studies, the ability to delay gratification might be one of the factors that play an important role in body mass control due to its influence on disinhibiting eating patterns [16]. The difficulties with inhibition of behavioral response may contribute to overeating, consuming additional snacks and extra food, and lead to weight gaining patterns [16]. Referring to our obtained results, children with excess weight proportionately more often ate or opened an egg (8.1% of children with excess weight in relation to 3.3% healthy participants) in the delay of gratification tasks. The calculated odds ratio suggested that children who preferred a reward granted immediately to the promise of a larger one later were more than twofold at the risk of being of excess weight. However, this association lost statistical significance after adjustment for the analysis for confounding factors such as socioeconomic status. Furthermore, there were no differences in BMI, LBMI, and FMI *z* scores between the groups of children who succeeded and who failed in the DGT. Unlike other research [15,28,43] and review [44], the failure in the DGT was not related to excess body mass in our study.

One of the most remarkable results of the DGT was the small number of children who ate or unpacked an egg-surprise (4.11%), which makes interpretation of analysis difficult. One of the possible explanations for this result may be the high SES of the investigated sample. The vast majority of children came from large cities and had at least one parent with a higher education level. In addition, most families described their material status as good enough. The link between DGT and body weight lost its significance precisely after including parental education as a cofounding factor. The parental role in shaping children’s self-regulation, including food intake, is possibly considerable [45]. Low-income and less educated parents are more likely to serve their children processed foods or give comfort foods to help children deal with stress. Their nutritional knowledge is also not as extensive as that of parents with a high SES. On the other hand, according to the meta-analysis of Protzko [46], over the last 50 years, children have increased their ability to delay gratification in general. According to the author, the possibilities of this improvement include, e.g., Flynn effects, earlier and better education, rising standards of living, or increased test awareness. Similarly, Carlson et al. [47] pointed to potential factors that may influence children’s improved performance on DG task: the Flynn effect, increased level of abstract thinking, increasing numbers of children enrolled in preschool, or changes in parenting.

As a confounding factor, the age of children was included as well. However, this variable was not significant in the logistic regression model. It is known that the ability to delay gratification improves with age [48]. The most improvements of the ability take place in early childhood up to 5–7 years of age, which is related to improved executive functioning in general and children’s theory of mind [47,49,50]. Wilson et al. [50] suggest that, after the age of 7, the *ceiling effect* occurs, and children gain the ability to delay reward. According to the researchers, gratification delaying based on the marshmallow test is reminiscent of an inverse U-shaped function (young and older people perform worse). As our sample was limited to primary school-aged children, the lack of differences in DGT according to age was not unusual. However, examining the ability to delay gratification over a couple days could probably similarly differentiate our results more—the younger the child, the greater tendency to wait longer for a bigger reward [48]. Therefore, an interesting perspective is to extend the study of the DG to *delay discounting* (DD), which is a simplified DGT procedure that tests a hypothetical delaying of a reward for a longer time than DG in the marshmallow test [49]. Furthermore, we did not measure children’s motivation in the DGT performance, such as attractiveness of stimulus, competitive indication, task perceived win–loss [51], nor did we ask them about their subjective level of difficulty in the task. This could diversify the results more. We have merely a behavioral assessment of the DGT performance.

According to the DGT–age–body mass relationship, with the exception of a few follow up studies on DGT tasks [24,27,52], there are limited data that can help to reveal the moderating effect of age on the association between EFs and obesity. According to these results, the inability to delay gratification in children was a significant predictor for weight gain or obesity in subsequent years of life. However, the available meta-analyses that explore the role of different developmental stages in EF–obesity studies come to varying conclusions. According to Pearce et al. [52], age might be important in the EF–obesity relationship, although not in the context of reward-related functions. Yang et al. [53] did not notice any moderating age effect on the hot self-regulation–body mass link.

Finally, in line with the dual process perspective [21], the absence of hunger might not have activated a *hot* state in participants. Hot visceral states (such as hunger or thirst) in the presence of unhealthy food are associated with a strong preference for immediate gratification. During a cold state, in the absence of temptation, an individual prefers healthy food and underestimates the future value of food in the presence of hunger (a cold–hot empathy gap) [21]. As we did not control the participants’ hunger/satiety level, we could not manipulate this variable in order to broaden the analysis.

### 4.2. Affective Decision-Making

Our analyses of the relationship between affective decision-making, body mass and adiposity were based on the results of the Hungry Donkey task. The literature suggests that such a relationship can be observed during childhood [16,23]. *Net score*, *learning rate*, and *learning curve* were taken into the analysis.

Surprisingly, the children with a net score >0.0 turned out to have slightly higher BMI and FMI *z* scores. However, the effect size was very small and suggested no significant differences. Furthermore, the ability to learn the game and to gain better results was not associated with either BMI status or body composition indicators. Better results in Trials 101–200 than in Trials 1–100 were observed for all participants, regardless of BMI status. Finally, the association of the net score with body mass index lost statistical significance after controlling for SES and mother’s BMI. Our results are consistent with a few other studies that did not find any significant association between impaired decision making and overweight children [15,16,18,54].

Nevertheless, the majority of research has shown the relationship between decision making and obesity to be significant [31]. The ability to learn a link between behavior, reward, and punishment, measured in the HDT task, is associated with features such as flexibility in decision-making [30]. Affective decision-making is related to the ability to regulate emotions and motivation, which seems to be of particular importance in the context of daily dietary choices and eating styles predisposing to excess weight [18]. It has been suggested that an inability to manage difficult emotions results in increased overeating tendencies [55] and the difficulty of giving up food with a high degree of tastiness [56]. However, food rewards in our study were not provided. It seems to be a research direction worth exploring direction.

In the study of Groppe and Elsner [30], children were told that they would win a marble when they collected 20 apples in order to increase motivational relevance. The perspective of win increases emotional arousal and activates hot executive functions. We did not offer any reward in our study procedure, as we assumed that the game itself should be motivational enough. It is possible that children needed a win perspective for activating affective decision-making.

The effect of the mother’s BMI suggests that other factors are more powerful determinants in shaping body weight than affective decision-making. Apart from the clear genetic influence, it is worth considering the phenomenon of obesity as a result of the entire system of family eating habits. We did not collect data on children’s eating styles and family dietary habits. However, the subject is worth exploring in the context of the EF–body mass relationship.

The children in our study demonstrated a performance of a task typical for their developmental age. It is possible that the relationship between affective decision-making and BMI is more pronounced in children with clinically relevant neurocognitive deficits, as in the case of attention-deficit/hyperactivity disorder [57] but might be unnoticeable in typically developed children. Moreover, while the association may be distinct in adolescents [58] or adults [59], in children, dietary habits and life-style are controlled by parents [60], thus, the link between EFs and body mass may not yet be visible. Cortese et al. [57] in his meta-analysis observed a link between ADHD and obesity in both children and adults; however, with an increasing strength of this link over time (40% increased prevalence of obesity in children and adolescents with ADHD and 70% increased prevalence of obesity in adults with ADHD in comparison to healthy controls). The association of development stage and EF–obesity link in typically developing children is another interesting research field worth exploring.

It is likely that, in children without serious mental and cognitive difficulties, minor impairments in EFs do not have such a strong effect on obesity development as in the case of children with neurodevelopmental disorders. The recent meta-analysis [53] indicates that the effects of executive functions on BMI or body weight are usually of small to medium size. They may remain hidden behind the larger effects of factors such as genetic determinations of metabolism and hunger/satiety regulation [61], socioeconomic factors or family life-style including diet, physical activity, and sedentary behaviors [62].

Therefore, a broader analysis of the relationship between food and satiety responsiveness and EFs may contribute to a better understanding of the phenomenon. This, in turn, may contribute to the development of practices strengthening the skills of food self-regulation and nutritional decision-making, which could have a positive impact on the prevention of overweight and obesity.

### 4.3. Limitations

There are some limitations that should be taken into consideration. The study is of a cross-sectional nature; thus, it was not possible to test the causality of the hot EF–BMI/FMI link. Furthermore, despite the fact we included potential confounders (age, SES) into the analyses, there remain other variables that we have not considered. There were no pre-experiment food consumption instructions for the children; thus, we did not control the possible influence of satiety and hunger on the children’s performance in the DGT. We did not take into consideration the children’s food preferences. Some of the authors asked them about their preferred snack, increasing the value of the reward [63]. Although, in our study, the egg-surprise (with a toy inside) was applied partly to enhance the stimulus effect; the number of children who did not complete the task with success was relatively small, which limits the possibilities of our conclusions on delaying gratification and body mass relationship. Next, although we used age as a confounding factor, we examined neither the developmental maturity of the participants nor their motivation to delay gratification and perceived level of difficulty in the task.

The low variation in the DGT scores prompts a further investigation to consider additional factors in our study that potentially influence better DGT performance. Such factors could include high levels of trust in the researchers and unfamiliar adult [64,65,66], positive parenting and parental sensitivity of examined children [67,68,69], high levels of religiosity in the family of origin [70,71], and higher IQs of participants [46]. In our study we did not control for these factors, which affects our capacity for drawing conclusions and interpretations.

Not controlling the food and water consumption before the study might have had a little impact on body composition analysis as well, although not clinically significant [72]. To control our analysis of the EF–BMI/body composition for age, we used age and sex adjusted *z* scores BMI, FMI and LBMI. Furthermore, we included age in a logistic regression analysis. These procedures allowed us to exclude the possible confounding effects of puberty onset in part of our sample, while data on pubertal development was not collected in this research. However, we did not evaluate the age moderation effect on EF–body mass status association. Finally, our sample included children without mental illness and clinical psychiatric diagnoses. Thus, strong EF deficits might have been difficult to observe. Nevertheless, the aim of our study was to investigate typically developed children and, thus, limit the impact of additional variables of a biological or psychosocial nature on both EFs and BMI/adiposity. These compounds should be explored in subsequent studies.

## 5. Conclusions

A relationship between the ability to delay gratification and affective decision-making with BMI status and body composition was previously found in children with clinically relevant EF deficits. Our study did not confirm this association in typically developed children age 6–12y. In this case, the effect of hot executive functions is likely to be small and masked by other genetic and environmental factors of well documented strong effects. Further studies should explore the association between the hot executive functions and adiposity, taking into the consideration the role of contributing factors, particularly food intake mechanisms and satiety responsiveness.

## Figures and Tables

**Figure 1 brainsci-11-00780-f001:**
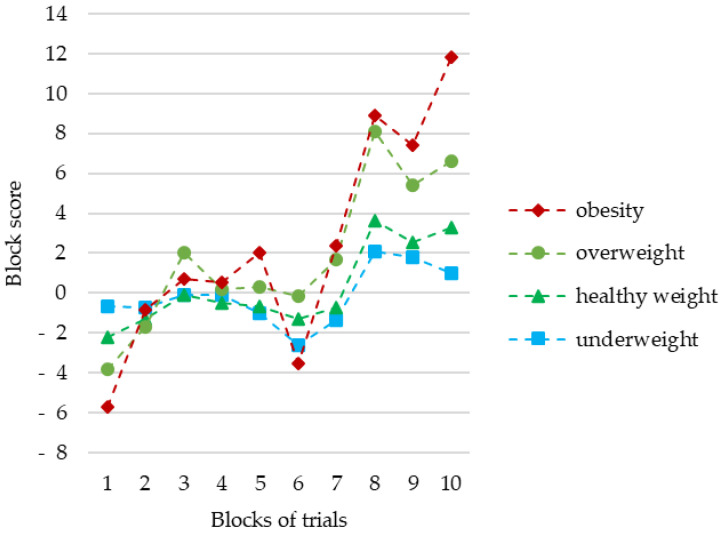
Mean scores by blocks of trials according to body mass status. No significant differences between the groups throughout the game (*p* = 0.161). The effect of learning observed in the whole sample (*p* = 0.005).

**Figure 2 brainsci-11-00780-f002:**
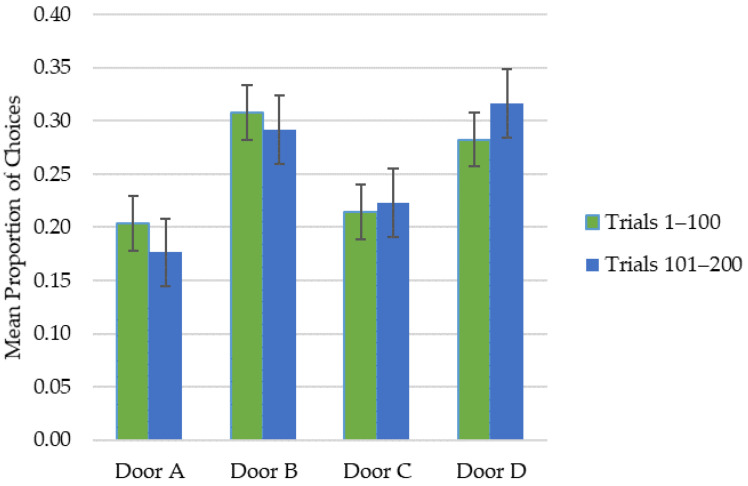
Mean proportion of choices by door for each 100-trial epoch in the whole sample. The differences between Trials 1–100 and Trials 101–200 are at significance *p* level < 0.001.

**Table 1 brainsci-11-00780-t001:** The differences in *z* scores BMI, FMI, and LBMI between children divided according to results of the delay of gratification task and the Hungry Donkey task. The *t*-test results.

DGT	Failure (n = 22)		Success (n = 511)					
	Mean	SD	Mean	SD	df	*t*	*p*	The Cohen’s *d* for Significant Differences
*z* scores BMI	0.82	1.36	0.32	1.21	529	−1.89	0.06	
*z* scores FMI	0.51	1.26	−0.02	0.97	525	−1.93	0.07	
*z* scores LBMI	0.19	1.13	0.00	0.99	525	−0.85	0.40	
HDT	impaired net score (n = 254)		net score > 0.0 (n = 253)					
	Mean	SD	Mean	SD		*t*	*p*	
*z* scores BMI	**0.22**	**1.22**	**0.47**	**1.23**	**503**	**−2.24**	**0.03**	**0.20**
*z* scores FMI	**−0.09**	**0.91**	**0.10**	**1.05**	**499**	**−2.12**	**0.03**	**0.19**
*z* scores LBMI	−0.05	0.98	0.05	0.99	499	−1.15	0.25	
	learning rate (−) (n = 204)		learning rate (+) (n = 303)					
	Mean	SD	Mean	SD		*t*	*p*	
*z* scores BMI	0.36	1.26	0.33	1.21	503	0.22	0.83	
*z* scores FMI	−0.01	1.01	0.02	0.97	499	−0.39	0.70	
*z* scores LBMI	0.08	1.01	−0.05	0.97	499	1.48	0.14	

DGT—delay of gratification task, HDT—Hungry Donkey task, BMI—body mass index, FMI—fat mass index, LBMI—lean body mass index, *t*—the t-Student test for equal variances or Welch’s *t* test for unequal variances, *p*—significance level, bold—significant difference (*p* < 0.05).

**Table 2 brainsci-11-00780-t002:** The results of logistic regression for excess body mass according to DGT results and cofounding factors.

Variable	B	SE(B)	Wald	df	*p*
*constant*	−3.241	1.018	10.137	1	0.001
DGT success	−0.841	0.474	3.142	1	0.076
Age	0.001	0.089	0.000	1	0.993
SES	−0.459	0.130	12.393	1	**0.007**
Mother’s BMI	−3.241	1.018	10.127	1	**0.000**

B—intercept; SE(B)—standard error; Wald—Wald chi-square test; df—degrees of freedom; *p*—significance level; bold—significant difference (*p* < 0.05), SES—socioeconomic status (parental education level).

**Table 3 brainsci-11-00780-t003:** The multiple linear regression result for BMI *z* score according to HDT outcome and cofounding factors. R^2^ =0.090, adjusted R^2^ = 0.084, SE = 1.18.

Variable	B	SE(B)	β	*t*	*p*
*constant*	−1.263	0.327		−3.866	**0.000**
Net score	0.401	0.236	0.074	1.696	0.091
SES	−0.132	0.064	−0.090	−2.041	**0.042**
Mother’s BMI	0.074	0.013	0.257	5.801	**0.000**

B—intercept; SE(B)—standard error; *t*—*t* test for linear regression; *p*—significance level; bold—significant difference (*p* < 0.05), SES—socioeconomic status (parental education level).

## Data Availability

The data that support the findings of the study are available from the corresponding author, [P.S.], upon reasonable request.

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
