# Peer review of "Do Hot Executive Functions Relate to BMI and Body Composition in School Age Children?"

_brainsci, 2021, doi:10.3390/brainsci11060780_

Round 1
Reviewer 1 Report
This study assesses gratification delay and decision making using behavioral tasks in children aged 6-12 years old. Despite having a relatively robust sample size, the authors found that initial associations between these tasks and obesity-related measurements were no longer significant when potential confounders (e.g., age and SES) were added in the model. The topic is certainly interesting, the manuscript is nicely written, and results are discussed appropriately.
An unexpected result for me was the lack of association between age and performance in the delay gratification task. Would it be possible that a score reflecting degree of developmental maturity (such as Tanner’s development stages) would have had an effect on gratification delay?
Author Response
Comment 1: This study assesses gratification delay and decision making using behavioral tasks in children aged 6-12 years old. Despite having a relatively robust sample size, the authors found that initial associations between these tasks and obesity-related measurements were no longer significant when potential confounders (e.g., age and SES) were added in the model. The topic is certainly interesting, the manuscript is nicely written, and results are discussed appropriately.
Answer: We thank the Reviewer sincerely for a favorable review and for a carefully summary of the manuscript. We are delighted that the Reviewer found the manuscript interesting.
Comment 2: An unexpected result for me was the lack of association between age and performance in the delay gratification task. Would it be possible that a score reflecting degree of developmental maturity (such as Tanner’s development stages) would have had an effect on gratification delay?
Answer: We thank the Reviewer greatly for this thoughtful suggestion. This is a highly valuable indication for us to include such an indicator in our future research. As we do not have such data for the study, it is therefore challenging for us to assess whether a score reflecting developmental maturity would have had the effect on gratification delay. However, to address the Reviewer’s suggestion, we have added the following to the discussion section:
[Line 317-330] It is known that the ability to delay gratification improves with age [48]. The most improvements of the ability take place in early childhood up to 5-7 years of age, what is related to improved executive functioning in general and children’s theory of mind [47, 49, 50]. Wilson et al [50]. suggest that after age of 7 the ceiling effect occurs, and children gain the ability to delay reward. According to the researchers, gratification delaying based on Marshmallow test is reminiscent of an inverse U-shaped function (young and older people perform worse). As our sample was limited to primary school-aged children, the lack of differences in DGT according to age was not unusual. However, examining the ability to delay gratification over a couple days could probably likewise differentiate our results more – the younger the child, the greater tendency to wait longer for a bigger reward [48]. Therefore, an interesting perspective is to extend the study of the DG to delay discounting (DD), which is a simplified DGT procedure that tests a hypothetical delaying of a reward for longer time than DG in Marshmallow test [49].
We have added the following to the limitations as well:
[Line 432-434] Next, although we used age as a cofounding factor, we examined neither the developmental maturity of the participants nor their motivation to delay gratification and perceived level of difficulty in the task.
We hope that we have met the Reviewer's expectations.
Reviewer 2 Report
In this worthwhile study, the researchers aim to unpick some of the myriad factors that contribute to childhood obesity. The study is extremely well written and clear. One minor issue centres on the use of the delay gratification task to quantify self-regulation. Although the ability to delay gratification is commonly associated with internal factors such as age and IQ, external factors such as socio-economic status, culture, family structure and even trust in the experimenter are also important predictors of the task. Although some surrogates of socio-economic status were included as confounding factors, nevertheless a more thorough discussion on the limitations of the DG task should be included.
Overall this is an excellent addition to the literature.
Author Response
Comment 1: In this worthwhile study, the researchers aim to unpick some of the myriad factors that contribute to childhood obesity. The study is extremely well written and clear. One minor issue centres on the use of the delay gratification task to quantify self-regulation. Although the ability to delay gratification is commonly associated with internal factors such as age and IQ, external factors such as socio-economic status, culture, family structure and even trust in the experimenter are also important predictors of the task. Although some surrogates of socio-economic status were included as confounding factors, nevertheless a more thorough discussion on the limitations of the DG task should be included.
Answer: We thank the Reviewer very much for the favorable words about the manuscript and for the Reviewer’s insightful comments on the DG task. According to these thoughtful suggestions, we have added the following to the discussion section:
[Line 330-334] Furthermore, we did not measure children’s motivation in the DGT performance, such as attractiveness of stimulus, competitive indication, task perceived win-loss [51], nor did we ask them about their subjective level of difficulty in the task. This could diversify the results more. We have merely a behavioral assessment of the DGT performance.
[Line 435-441] The low variation in the DGT scores prompts a further investigation to consider additional factors in our study that potentially influence better DGT performance. Such factors could include high levels of trust in the researchers and unfamiliar adult [64-66], positive parenting and parental sensitivity of examined children [67-69], high levels of religiosity in the family of origin [70, 71], and higher IQs of participants [46]. In our study we did not control for these factors, which affects our capacity for drawing conclusions and interpretations.
We hope that we have met the Reviewer’s expectations.
Comment 2: Overall this is an excellent addition to the literature.
Answer: Many thanks to the Reviewer for the words of appreciation. It is of great value to us.